# Application of *tris*-(4,7-Diphenyl-1,10 phenanthroline)ruthenium(II) Dichloride to Detection of Microorganisms in Pharmaceutical Products [note 1]

**DOI:** 10.3390/ph16060856

**Published:** 2023-06-08

**Authors:** Rafał Hałasa, Katarzyna Turecka, Magdalena Smoktunowicz, Urszula Mizerska, Czesława Orlewska

**Affiliations:** 1Department of Pharmaceutical Microbiology, Faculty of Pharmacy, Medical University of Gdańsk, al. J. Hallera 107, 80-416 Gdańsk, Poland; katarzyna.turecka@gumed.edu.pl (K.T.); magdalena.smoktunowicz@gumed.edu.pl (M.S.); 2Department of Polymeric Nano-Materials, Centre of Molecular and Macromolecular Studies, Polish Academy of Sciences, ul. Sienkiewicza 112, 90-363 Lodz, Poland; urszula.mizerska@cbmm.lodz.pl; 3Department of Organic Chemistry, Faculty of Pharmacy, Medical University of Gdańsk, al. J. Hallera 107, 80-416 Gdańsk, Poland; czeslawa.orlewska@gumed.edu.pl

**Keywords:** oxygen sensor, fluorescence optical respirometry, plant extracts, sterile pharmaceutical products, non-sterile pharmaceutical products, European Pharmacopoeia, antimicrobial activity

## Abstract

*tris*-[(4,7-diphenyl-1,10-phenanthroline)ruthenium(II)] dichloride (Ru(DPP)_3_Cl_2_), a fluorescent sensor which is sensitive to the amount of oxygen in the sample, was applied using the fluorescent optical respirometry (FOR) technique. The oxygen in the samples quenches the fluorescence. The fluorescence intensity depends on the metabolic rate of the viable microorganisms. The effect of DMSO and plant extracts on bacteria was determined by FOR. It was shown that the MIC values obtained by FOR were consistent with the results of the MIC determinations using the method of serial dilutions; at the same time, the effects of concentrations lower than the growth-inhibitory concentrations on microbial cells were demonstrated. The FOR method enables the detection of multiplying bacteria in sterile and non-sterile pharmaceutical preparations in real time, which significantly shortens the time required to obtain results and allows the introduction of repair processes in the production. This method also allows for quick, unambiguous detection and the counting of the viable cells of aerobic microorganisms in non-sterile pharmaceuticals.

## 1. Introduction

The determination of microbial numbers plays a vital role in all industrial biotechnological processes that utilize fermentation technologies. The measurement of bacterial and fungi biomass provides the opportunity to evaluate microbial growth, synthesis efficiency, speed, and bioprocess balance. The evaluation of the quality of pharmaceutical products needs selective, sensitive, and fast methods to detect a small number of viable bacterial cells. The search for new antibacterial compounds requires the testing of many compounds on a large number of microorganisms.

The European Pharmacopoeia contains two areas of research: the search for microorganisms in sterile products and the determination of the number of microorganisms in non-sterile products [1].

The first area (Ph. Eur. 2.6.1), which is for parenteral drug products that are required to be free from any viable microorganisms, includes methods based on the aseptic inoculation of samples into liquid media and incubation for up to 14 days. High-turbidity bacterial density (over 10^7^ CFU/mL) (CFU = colony-forming units) indicates the presence of microorganisms. Two methods are used: membrane filtration and direct inoculation [1].

The second area (Ph. Eur. 2.6.12 and 2.6.13) of research is for oral and topical products, for which there are strict guidelines limiting the number and types of acceptable microorganisms. In this area of microbial enumeration, total aerobic microbial count (TAMC), analysis, total yeast and mold count (TYMC) analysis, and tests for specified microorganisms are employed. The harmonized methods include the counting of bacterial colony-forming units on agar plate membrane filters and the application of the “most probable number”; these are the basic methods used to determine viable bacterial cells. In the first two methods mentioned above, a long incubation time (1–3 days) is required, colonies may be formed by several related species of bacteria, and full identification takes up to seven days, whereas with the use of the “most probable number” method an approximate number of bacteria can be detected in a diluted test sample by measuring turbidity after incubation. The techniques require clear, dispersed samples without inhibitory factors [1,2].

Pharmacopeia also permits the detection of microorganisms by alternative methods (Ph. Eur. 5.1.6) [1].

Turbidimetry and nephelometry are non-invasive, cheap, rapid, and automated methods for measuring bacterial growth. Nevertheless, these methods require dispersed samples and high bacterial density (over 10^7^ CFU/mL) [3]. In turn, the nucleic acids and fatty acid detection assays are characterized by high sensitivity (about 10 CFU/mL) and short detection time (6 to 18 h). Unfortunately, several steps of cell detection, specialized equipment, and trained personnel are required. Moreover, distinguishing between living and dead cells is also difficult [1,3].

Bioluminescence and chemiluminescence, among others, are the most important methods for detecting bacterial pathogens. In the industry, luminescence measurements allow the continuous monitoring of technological processes. They use special systems such as (*lux*) *Vibrio harveyi* and *Vibrio fischeri* in bacteria or (*luc*) in insects. These systems have been adapted for the detection of microorganisms. The method has been commercialized, and it is rapid and automated. The limitations of this technique are the use of a bacterial strain with special growth requirements and test samples that may be turbid or stained or contain chlorine [4,5].

Techniques such as those using surface plasmon resonance (SPR), a Coulter counter, electrorotation assay, or electrochemical impedance spectroscopy allow real-time density measurements of samples. Unfortunately, these samples must contain microorganisms with a density of 5 × 10^4^ to >10^8^ CFU/mL, and there is no differentiation between dead and living cells. The apparatus is quite complicated and requires properly trained staff [4].

In the face of the spreading increase in the antibiotic resistance of microorganisms, new chemical compounds with antimicrobial properties are being sought. Currently, many research centers are trying to obtain active compounds from medicinal plants. The method of serial dilutions and the diffusion method of paper discs are commonly used for testing the potency of chemical compounds. The serial dilution method, which is used to determine the minimum inhibitory concentration (MIC), can be applied to test large numbers of samples against several strains of bacteria [6,7,8,9]. Unfortunately, the physical properties of the sample affect the results and assay time, and these are only some of the disadvantages of the method. However, it is considered to be a standard means of evaluating the antimicrobial activity of tested compounds.

Colorimetry and radiometry, the subsequent bacterial detection methods, focus on monitoring the metabolism as a marker of the presence of living cells. The main sources of information about bacterial growth are the detection of O_2_ consumption, CO_2_, or ATP release. The detection of ATP is performed by ATP bioluminescence assay using a luciferase enzyme. The method is fast and inexpensive but can give false positive results. Radiometric methods using radioactive ^14^C are commonly used to detect *Mycobacterium tuberculosis*, food-borne pathogens, and bacteria from water samples. Sensors responding to radioactivity change are located at the bottom of the incubation vessel (bottle, test tube, or titer well) [10,11]. The changes in the sensor are recorded by a special detection system. This method allows the detection of a single bacterial cell in 8–16 h [2]. However, specialized equipment and specific sensors generate high costs. Similarly, colorimetric methods based on CO_2_-sensitive sensors work by changing the color of the indicator in the presence of the mentioned compound. This method has been used in the Bact/Alert company Organon Technika and ESP Microbial Detection System (AccuMed International) [12,13].

The analysis of the microbial growth in cultures also uses another method of optical respirometry [14,15,16,17,18,19]. This technique is based on the analysis of the fluorescence of an oxygen-sensitive sensor. Molecular oxygen is a fluorescence quencher; the growth of microorganisms in culture is accompanied by oxygen consumption, which affects the intensity of the fluorescence in the sample. Thus, by analyzing variations in the sample’s fluorescence intensity, the metabolic activity of microorganisms can be monitored. Fluorescence optical respirometry methods have been successfully applied to track the oxygen respiration of bacteria, yeast, and mammalian cells, as well as to detect the action of some antibiotics and chemicals compounds. Attempts are underway to introduce biosensors for the qualitative control of the food production process, as well as for checking the suitability of food for consumption.

A sensor coating cocktail containing polymeric microparticles impregnated with Pt(II)-tetrabenzoporphyrin (PtBP) dye containing hydrogel (binder) is being tested as a sensor of the microbiological quality of food products. The suitability of the quality control of the food production process was checked [16,20,21,22].

In this publication, we are the first to present the application of the method using *tris*-[(4,7-diphenyl-1,10-phenanthroline)ruthenium(II)] dichloride (Ru(DPP)_3_Cl_2_) for the rapid, real-time measurement of growth and the calculation of living aerobic microorganism cells in sterile and non-sterile pharmaceutical products. We are the first to show the usefulness of the method to study the mechanisms of action of the plant extracts against bacterial culture. As reference methods, we used the serial dilution method in broth and colony counts on agar plates.

## 2. Results and Discussion

### 2.1. Effect of Dimethyl Sulfoxide against Bacteria

In the studies of the antimicrobial activity of chemical compounds, when the water-insoluble compounds are tested, dimethyl sulphoxide (DMSO) is often used as a solvent [23,24,25,26]. It is widely known that DMSO has a toxic effect on microorganisms at higher concentrations. Therefore, it is important to determine the concentrations in which DMSO can be used to prepare samples for the evaluation of the MICs of newly synthesized chemicals. The mechanism of DMSO toxicity has not yet been revealed, but it is known that it is a “scavenger” of hydroxyl radicals and that it abolishes the effect of menadione, which is affected by the generation of peroxides [27]. The possible application of DMSO as an active substance in medicines has been reported [28].

The results of studies analyzing the effects of different concentrations of DMSO against *Salmonella enterica*, *Staphylococcus aureus*, *Escherichia coli*, and *Bacillus subtilis* are shown in Figure 1a–d.

Figure 1 shows that DMSO inhibits the growth of bacterial cultures at concentrations higher than 13% (*vol*/*vol*) for *S. enterica*, *E. coli*, *B. subtilis* and 10% (*vol*/*vol*) for *S. aureus*. In the presence of 2, 5, 8, 10, and 13% DMSO, the curves of fluorescence intensity obtained for the culture retain the same sigmoidal shape, but they are increasingly shifted in time when the higher concentration of DMSO is used. Thus, the higher the concentration of DMSO, the greater the retardation that occurs. Worthy of note are the curves assigned to the *S. enterica* control culture (Figure 1c); the shapes of the curves are very sharp and do not reach the values achieved by the other curves. This may be a result of a very rapid oxygen consumption in the sample—rapid growth of bacteria. Thus, the obtained fluorescence values are maximal for the amount of foreign sensor in these wells. The curves assigned to the control *E. coli* (Figure 1a) culture, the same culture treated with 5% DMSO, and the *B. subtilis* cells treated with 2 and 10% DMSO (Figure 1d) have significantly lower maximum values than the other curves. The differences in fluorescence intensity may result from different amounts of sensor in the individual wells. The presence of such differences is due to the difficulty in dispensing the sensor–silica mixture in Davosil gel due to the viscosity and thick consistency.

The dependence between the concentration of DMSO and the time at which the fluorescence intensity reached half the maximum value is shown for *E. coli* in Figure 2.

Figure 2 presents the linear dependence of the culture-specific parameter against the DMSO concentration. It can be seen that with the increasing DMSO concentration in the suspension of bacteria, the time it takes for the bacteria to reach half their fluorescence intensity is proportionally extended.

The observation is also consistent with the MIC value for tested bacteria determined by the serial dilution method.

The MIC values for the DMSO obtained by optical fluorescence respirometry were consistent with the MIC values achieved by the serial dilution method. (Figure 3).

The analysis of the inhibitory effect of DMSO on the bacteria when using the traditional method and when using fluorescence optical respirometry implied that both methods gave the same MIC value. However, the FOR technique supplied much more information compared with the serial dilution method, particularly with regard to the toxic effect of low concentrations of DMSO on bacteria. Moreover, the FOR method significantly reduced the time of the analysis; the MIC values were readable after four hours. In contrast, by increasing the initial number of bacterial cells in the culture, this result could be obtained with a significant reduction in the test time to 1–2 h [16].

Our respirometric assay results confirmed those obtained by O’Mahony and Papkovsky [28], where DMSO inhibited the growth and metabolism of *E. coli* at 15% DMSO. The authors showed that the turbidimetric assay inhibited only bacterial growth but not metabolism. The final concentration used in the tests of *E. coli* was ~10^8^ CFU/mL for O’Mahony and Papkovsky 2006, but our results were obtained from an initial concentration of 10^5^ CFU/mL. The researchers used water-soluble Pt–porphyrin-based type A65N-1 sensor (Luxcel Biosciences, Cork, Ireland) and tested different fluorescence readers.

### 2.2. Effect of Plant Extracts against Bacteria

Despite the undeniable achievements in the field of microbiology, morbidity and mortality are increasing worldwide because of infectious diseases caused by increasingly common drug-resistant strains of pathogenic bacteria and fungi. The necessity to develop more effective and non-toxic antimicrobials directs the attention of scientists towards plant-derived compounds as a source of new antimicrobial drugs. Therefore, preliminary in vitro screening for the antimicrobial activity of plant extracts can help to select those with notable activity to become sources of new compounds for further clinical trials [29]. The diffusion and serial dilution methods allow the determination of the activity of the extracts. Serial dilution allows the determination of the minimum inhibitory concentration (MIC) [30,31].

Examples of the use of FOR to study the effect of extracts against bacteria are presented in Figure 4a–f.

The analysis of the effect of extracts from *Lonicera caerulea* var. *edulis* ‘Wojtek’ fruits (Figure 4) on the studied bacteria showed that concentrations of 41.625 mg/mL inhibited the growth of *Enterococcus hirae*, *Escherichia coli*, *Proteus vulgaris*, and *Pseudomonas aeruginosa*, whereas the concentrations greater than 10.4 mg/mL were effective against *Staphylococcus aureus* and *Staphylococcus epidermidis.* The results obtained by FOR were in accordance with the results of the activity determination made by the traditional method in the publication by Kula et al. [32].

The analysis of the curves presented in Figure 5 show that the extracts from *Rubus idaeus* shoots are active against *Klebsiella pneumoanie* and *Corynebacterium ditphtheriae* bacteria at concentrations higher than 41.625 mg/mL and 0.625 mg/mL, respectively. These results are consistent with those obtained by the traditional method in the publication by Krauze-Baranowska et al. [33]. Plant extracts are multicomponent compositions, including aromatic compounds recognized as secondary metabolites of plants (such as anthocyanins and flavonoids), which give them various physical–chemical properties, e.g., color and turbidity. These properties sometimes make it difficult to read the results correctly. Based on the results, it can be said that the extract components do not interfere with the fluorescence of the sensor, and the results obtained during the measurements were consistent with those determined by the method of serial dilutions. Analyzing the curves in Figure 4, it can be seen that concentrations lower than the MIC affect the growth of bacteria. This is particularly evident in the case of *S. aureus*, *S. epiderminis*, where a significant shift in the growth curve of the bacteria in the presence of 5.2 mg/mL extract compared to the control occurs. Figure 4b,e] show the strong bacteriostatic effect of the extracts from *Lonicera caerulea* var. *edulis* ‘Wojtek’ fruits against staphylococcal cells at concentrations below the MIC. The shift in the pattern curves is the result of the inhibitory effect of extract concentrations lower than the MIC on the cell. The slowdown in cell division was significant and ranged from 4.5 to over 7 h compared to the controls. Concentrations lower than the MIC also affected Gram-negative bacteria, but the bacteriostatic effect was not as significant (especially at the lowest concentrations) as it was in the case of staphylococci. The noticeable differences in sensitivity may be due to differences in the structure of the cell wall of the Gram-negative and Gram-positive bacteria. It should be added that the method turned out to be useful for the study of slow-growing *Corynebacterium diphtheriae*. Small sample volumes made it possible to quickly capture changes in bacterial metabolism, much faster than in the traditional method.

### 2.3. FOR Method Suitability Test for the Detection of Aerobic Bacteria in Non-Sterile Pharmaceutical Products

The FOR validation was based on monograph 2.6.12 of the European Pharmacopoeia 10.0, in the chapters on the usefulness of the method [1]. The following strains of microorganisms were used in the research: *Staphylococcus aureus* ATCC6538, *Bacillus subtilis* ATCC 663, *Escherichia coli* ATCC 8739, *Salmonella enterica* ATCC 13076, *Pseudomonas aeruginosa* ATCC 9027, and *Candida albicans* ATCC 10231.

The recommendations of the microbiological examination of non-sterile products are associated with microbial enumeration, in accordance with the method suitability test of Ph. Eur. [31]. The sample preparation depends on the composition of the product. Water-soluble products are diluted (usually a 1 in 10 dilution is prepared) with casein soya bean digest broth. Further dilutions, if necessary, are prepared with the same diluent. Next, a sufficient volume of the microbial suspension is added to the prepared sample and to the control (with no test material included) to obtain an inoculum of no more than 100 CFU/mL. The volume of the inoculum suspension should not exceed 1 percent of the diluted product volume. To demonstrate acceptable microbial recovery from the product, the lowest possible dilution factor of the prepared sample must be used for the test. In order to determinate the total aerobic microbial count (TAMC), casein soya bean digest agar was used. According to the Ph. Eur., plate counting methods (the pour-plate and surface-spread methods) should be performed in at least two replicates for each medium and the average number of results used. The recommended incubation time is no more than 3 days at 30–35 °C for bacteria and no more than 5 days at 20–25 °C for yeast and fungi.

The following products were used for testing: the ear drops “Ototalgin”, manufactured by Farmina sp. zoo, Kraków; the cough syrup “Dexa Pico”, manufactured by Herbapol-Lublin, Lublin; Injection Natrii Chlorati Isotonica (INCI) in a volume of 10 mL, manufactured by Polpharma, Starogard Gdański; and the diet supplement herbal syrup “Gardlox”, manufactured by S-lab sp. zoo, Mirków, Poland.

The results are shown in Figure 6a–f.

The above results showed the usefulness of FOR for the detection of microorganisms in non-sterile pharmaceutical products.: in the ear drops “Ototalgin”, in the “Dexa Pico” cough syrup, and in the dietary supplement herbal syrup “Gardlox” (Figure 6). The fluorescence intensity curves, obtained for bacterial cultures in the presence of non-sterile product concentrations, had the same sigmoidal character as the control, but in the presence of increasing product concentrations, they were increasingly shifted in time. It is necessary to neutralize the preservatives in the tested products; otherwise, they may inhibit the growth of microorganisms.

FOR significantly shortened the time of the studies from 3 days for bacteria and 5 days for yeasts to 10 and 15 h, respectively. Based on the tested preparations, it can be concluded that the preparation itself does not affect the level of fluorescence of the sensor, nor does it affect the signal emitted by the sensor during the growth of bacteria or yeast in the tested samples. Shortening the time to obtain results with the FOR method also allows the reduction in the time to obtain results in the purity tests of the pharmaceutical preparations, which will also speed up the possibility of releasing the product to the market. The usefulness of the FOR method in maintenance tests should also be checked. There is a noticeable shift in the curves corresponding to the most concentrated preparations in relation to the more diluted ones and to the bacteria themselves in the broth. This may be the result of the weaker neutralization of the active compounds or the influence of other pharmaceutical ingredients. The effect cannot be observed in the method of detection recommended by the Ph. Eur. These results confirm the high sensitivity of the method and draw attention to the proper selection of the neutralizer, which indirectly affects the rate of growth of the microorganisms. To determine the number of aerobic microorganisms in the tested samples, the relationship presented in monograph [16] in Figure 2a,b should be used. In the mentioned work, experiments were conducted at the temperature of 37 °C for *Escherichia coli* ATCC 8739, but these dependencies may apply to the detection of the number of microorganisms in the volume in the wells, as well as in other volumes in pharmaceutical preparations, when using the growth of standard strains under the same incubation conditions as the samples as a reference.

In our earlier publication [17], we presented the graphs in Figure 2a,b. The presented results refer to the determination of the number of viable cells of *E. coli* at 37 °C in wells, but similar relationships can be drawn up for other aerobic bacteria. A prerequisite for obtaining results is the presence of a control in the test—a reference strain without a pharmaceutical product.

### 2.4. FOR Method Suitability Test for the Detection of Aerobic Bacteria in Sterile Pharmaceutical Products

The FOR validation was based on monograph 2.6.1 of the European Pharmacopoeia 10.0, in the chapters on the usefulness of the method [1]. The following strains of microorganisms were used in the experiments: *Staphylococcus aureus* ATCC6538, *Bacillus subtilis* ATCC 663, *Escherichia coli* ATCC 8739, *Salmonella enterica* ATCC 13076, *Pseudomonas aeruginosa* ATCC9027, and *Candida albicans* ATCC 10231.

The following products were used for testing: the eye drops “Polcrom”, manufactured by Polfa Warszawa and Natrii Chlorati Isotonica (INCI) at a volume of 10 mL, manufactured by Polpharma, Starogard Gdański. All the steps of the experiments were carried out in accordance with the sterility studies of the method suitability test of the Ph. Eur. [1] recommendations.

Thus, after transferring the tested samples to the culture medium, an inoculum of no more than 100 CFU/mL needed to be added to the medium. The positive control consisted of bacteria and fungi grown in the medium for no longer than 5 days.

The results are shown in Figure 7a–f.

Figure 7 shows the graphs of the bacterial growth in the sterile pharmaceutical products with a characteristic sigmoid shape. Based on the curves, we demonstrated the usefulness of FOR in detecting microorganisms in sterile pharmaceutical products after only 3 h of measurement, even with a very low number of microorganisms. The longest time needed to detect microbes was 12 h (10^2^ CFU/mL). However, the time of microorganism detection in the product was much shorter compared to the standard methods, where the recommended incubation time of samples is 5 days (5 days). Next, it is necessary to develop a method for the rapid detection of anaerobic bacteria in sterile products.

## 3. Materials and Methods

### 3.1. Materials

Oxygen-sensitive sensor: the *tris*-[(4,7-diphenyl-1,10-phenanthroline)ruthenium (II)] chloride (Ru(DPP)_3_Cl_2_) sensor was prepared according to the method of Watts and Crosby [34], with modifications. The following were used: mineral oil (Sigma-Aldrich, Burlington, MA, USA); Silica gel, Davisil™, grade 633, 200–425 mesh, 60A, 90%, (Sigma-Aldrich); 4,7-diphenyl-1,10-phenanthroline, C_24_H_16_N_2_ (Sigma-Aldrich); DMSO-(CH_3_)_2_SO-dimethyl sulfoxide 100% (Reachim); and ruthenium chloride (III) (Sigma-Aldrich). Plant extracts from *Lonicera caerulea* var. *edulis* ‘Wojtek’ flowers and *Rubus idaeus* shoots were prepared and their composition was determined at the Department of Pharmacognosy with Medicinal Plant Garden, Faculty of Pharmacy, Medical University of Gdańsk, as described by Krauze-Baranowska et al. [30]. The pharmaceutical products were commercially available in pharmacies: the ear drops “Ototalgin”, produced by the company Farmina sp. z o.o., Kraków; the eye drops “Polcrom”, produced by the company Polfa Warszawa, Warszawa; the cough syrup “Dexa Pico”, produced by the company Herbapol-Lublin, Lublin; Injection Natrii Chlorati Isotonica in a volume of 10 mL, produced by Polpharma, Starogard Gdański; and the diet supplement herbal syrup “Gardlox”, produced by the company S-lab sp. z o.o., Mirków, Poland.

### 3.2. Microorganisms Strains and Growth Conditions

Gram-positive bacteria: *Enterococcus hirae* ATCC 1051, *Staphylococcus epidermidis* ATCC 14990, *Staphylococcus aureus* ATCC 6538, *Bacillus subtilis* ATCC 6633, and *Corynebacterium diphtheriae*; Gram-negative bacteria: *Escherichia coli* ATCC 8739, *Salmonella enterica* ATCC 13076, *Klebsiella pneumoniae* (clinical isolate), *Proteus vulgaris* NTCT 4635, and *Pseudomonas aeruginosa* ATCC 9027; yeast: *Candida albicans* ATCC 10231. The reference strains originated from the LGC standards. The clinical strains of bacteria from the Department of Pharmaceutical Microbiology collection were used. The antibacterial activity of the chemical compounds and extracts were performed in Mueller–Hinton broth (MH cation-adjusted, Becton Dickinson) (for most of the tested strains) in an aerobic atmosphere at 37 °C for 48 h. *Corynebacterium diphtheriae* was grown in brain–heart infusion broth (BHI, Becton Dickinson), supplemented with 10% bovine serum in an aerobic atmosphere at 37 °C for 48 h. *Enterococcus hirae* ATCC 1051 was cultured in brain–heart infusion broth (BHI, Becton Dickinson) in an aerobic atmosphere at 37 °C for 48 h. In order to determine the microbial viability, MH or BHI agar plates were used. Overnight microbial cultures were diluted in geometric progression with Mueller–Hinton broth or BHI broth. Then, 100 µL of each dilution was inoculated in agar plates and incubated at 37 °C for 24 h. After incubation, the colonies of bacteria were counted and the CFU/mL was determined.

The FOR method suitability test for the detection of aerobic bacteria in pharmaceutical products, the media, and the incubation conditions are contained in the guidelines of the European Pharmacopoeia, monographs 2.6.1 and 2.6.12, in the chapters on the usefulness of the method. The following were used: tryptic soya broth (TSB, Becton Dickinson), temperature 30–35 °C; bacteria: *Staphylococcus aureus* ATCC 6538, *Bacillus subtilis* ATCC 663, *Escherichia coli* ATCC 8739, *Salmonella enterica* ATCC 13076, and *Pseudomonas aeruginosa* ATCC 9027; and yeast: *Candida albicans* ATCC 10231.

### 3.3. Synthesis of tris-(4,7-Diphenyl-1,10-phenanthroline)ruthenium(II) Chloride (Ru(DPP)_3_Cl_2_) Biosensor and Coating the Walls of the 96-Well Microtiter Plates

One hundred and four milligrams of RuCl_3_ and 0.25 mL of water were mixed with 3 mL ethylene glycol. The salt was dissolved at 120 °C. Next, 500 mg of 4,7-diphenyl-1,10-phenanthroline was added and irradiated in a microwave reactor for 5 min. Subsequently, the reaction product was cooled to room temperature and mixed with chloroform (30 mL), then washed with a saturated solution of sodium chloride (40 mL). The organic layer was separated and evaporated to dryness, and the residue was recrystallized from an ethanol:water (2:1) mixture. The planned compound was achieved (475 mg)*. Anal.Calc.* for C_72_H_48_Cl_2_N_6_Rux 5H_2_O: C, 68.67; H, 4.64; N, 6.67. Found: C 65.90; H 4.25; N 6.10.

Ru(DPP)_3_Cl_2_ was mixed with Davisil™ silica gel by evaporation (0.9 mg Ru(DPP)_3_Cl_2_/g silica gel) and embedded in silicone rubber Lactite^®^ NuvaSil^®^ 5091 (2% *w*/*w*). Next, 2 drops were added onto the wells of the microtiter plates (MedLabor Greiner Bio-One company, Frickenhausen, Germany); then, the plate was incubated in a moist chamber for 2–3 days at 37 °C and used for tests.

Measurements were made with a Tecan Infinite 200PRO, but any 96-well plate reader capable of measuring fluorescence could have been used for this purpose.

### 3.4. Determination the Antimicrobial Properties of Compounds Using the Microbroth Dilution Method

The MIC and MBC of the tested compounds were performed according to the procedure described by Hałasa et al. [17], with modifications. The final concentrations of the compounds used for the testing of the antimicrobial activity ranged as follows: DMSO—from 20 to 2%; extract from *Lonicera caerulea* var. *edulis* ‘Wojtek’ flowers and from *Rubus idaeus* shoots—from 166 to 0.08 mg/mL.

### 3.5. Measurement of the Effect of Selected Chemicals on Bacteria by Fluorescence Optical Respirometry

Proper solution (80 µL) was added to the wells of the microtiter plate coated with oxygen biosensor: DMSO (40, 36, 30, 26, 20, 16, 10, 4%); plant extracts form *Lonicera caerulea* var. *edulis* ‘Wojtek’ fruits (166, 83.25, 41.625, 20.8, 10.4, 5.2 mg/mL); and *Rubus idaeus* shoots (166, 83.25, 41.625, 20.8, 10.4, 5.2 mg/mL or 16.6, 8.3, 4.15, 2.075, 1, 0.6 mg/mL). Eighty microliters of the bacteria suspension (10^6^ CFU/mL) was introduced to each well. Mineral oil (60 µL) was applied on the surface of the test samples, and the plate was placed in a Tecan Infinite 200PRO. All experiments were performed in triplicate.

### 3.6. FOR Method Suitability Test for the Detection of Aerobic Bacteria in Sterile and Non-Sterile Pharmaceutical Products

The FOR validation was based on monographs 2.6.1 and 2.6.12 of the European Pharmacopoeia 10.0, in the chapters on the usefulness of the method [1]. The products used for testing were as follows: the ear drops “Ototalgin”, produced by the company Farmina sp. z o.o., Kraków; the eye drops “Polcrom”, produced by the company Polfa Warszawa, Warszawa; the cough syrup “Dexa Pico”, produced by the company Herbapol-Lublin, Lublin; Injection Natrii Chlorati Isotonica in a volume of 10 mL, produced by Polpharma, Starogard Gdański; the diet supplement herbal syrup “Gardlox”, produced by the company S-lab sp zoo, Mirków, Poland; and tryptic soya broth (TSB, Becton Dickinson), temperature 30–35 °C. The bacteria used were: *Staphylococcus aureus* ATCC 6538, *Bacillus subtilis* ATCC663, *Escherichia coli* ATCC 8739, *Salmonella enterica* ATCC 13076, *Pseudomonas aeruginosa* ATCC 9027; and *Candida albicans* ATCC 10231.

## 4. Conclusions

The aim of this work was to demonstrate the usefulness of optical fluorescence respirometry for detecting and counting microorganisms in various pharmaceutical products, regardless of their composition and assigned application. By examining the pharmaceutical preparations—especially syrups and extracts—we showed that despite the complex composition the obtained results were reliable and unambiguous and that the composition of the product did not affect the florescence signal.

The DMSO studies showed once again that FOR is a quick and objective screening method for determining the antimicrobial properties of the chemical compounds used as solvents for active pharmaceutical substances. The analysis of the inhibitory effect of DMSO on the bacteria, conducted using the traditional method and fluorescence optical respirometry, gave the same MIC values. However, FOR makes it possible to see the effect of concentrations below the MIC on bacterial cells. Using FOR, we were able to reduce the time of examinations, even to as low as 4 h. To the best of our knowledge, we demonstrated here for the first time the possibility of using FOR to study the effects of plant extracts with compositions of various ingredients on bacteria. Based on the measurements using FOR, we were the first to elucidate the effect of concentrations lower than the growth-inhibitory concentrations (MIC) on microbial cells. Once again, we showed that our method, with its objectivity of determinations, significantly shortened the time to obtain results. We were also the first to show the possibility of using the FOR method for research on *Corynebacterium diphtheriae*. The suitability of the method for research on *Mycobacterium tuberculosis* should be checked.

We were the first to demonstrate the ability to detect aerobic microbes in selected non-sterile pharmaceutical products. We showed that by using FOR the presence of aerobic microorganisms can be detected 3 h after setting the samples, while the Ph. Eur. guidelines recommend reading the results after 3 days.

We were the first to demonstrate the possibility of using the FOR method to detect aerobic bacteria in selected sterile pharmaceutical products. We showed that by using FOR the presence of aerobic microorganisms can be detected 3 h after setting the samples, while the Ph. Eur. guidelines recommend reading the results after 5 days. Once again, we showed that our method, with its objectivity of determinations, significantly shortens the time to obtain results. Reducing the time needed for microorganism detection and the objectivity of the method will shorten the time needed for the detection of possible biological contaminants in sterile drugs.

## Figures and Tables

**Figure 1 pharmaceuticals-16-00856-f001:**
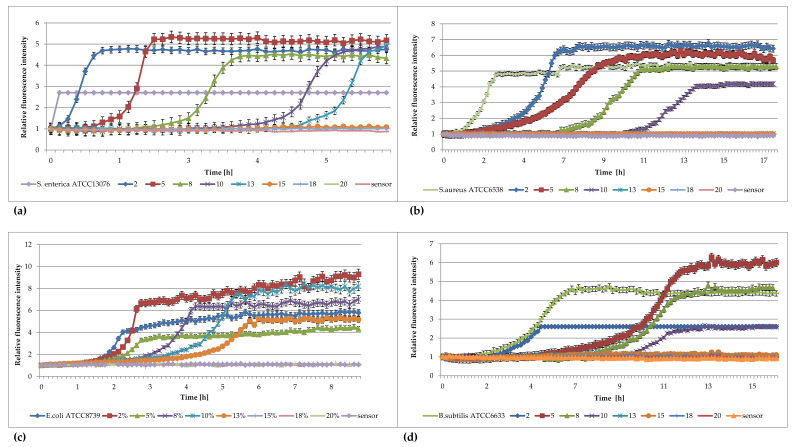
Plots of the relative fluorescence intensity of (**a**) *Salmonella enterica* ATCC 13076, (**b**) *Staphylococcus aureus* ATCC 6538, (**c**) *Escherichia coli* ATCC 8739, and (**d**) *Bacillus subtilis* ATCC 6633 cultures against time for different concentrations of DMSO (%).

**Figure 2 pharmaceuticals-16-00856-f002:**
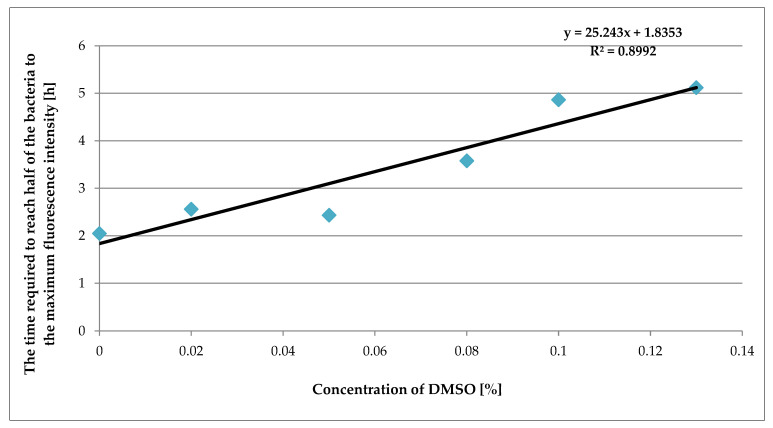
A plot of the time required to reach half of the maximum value of fluorescence intensity versus the concentration of DMSO.

**Figure 3 pharmaceuticals-16-00856-f003:**
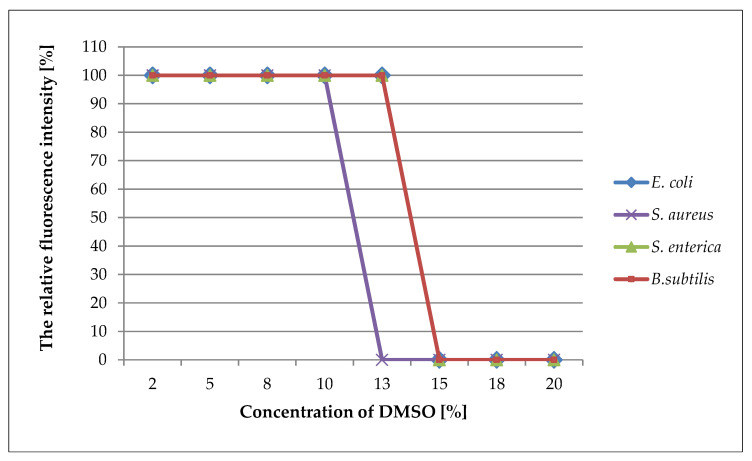
Plots of the maximum relative fluorescence values obtained by bacterial cultures versus the concentration of DMSO. The maximum value of fluorescence intensity achieved by the growing bacterial cultures was taken as 100%.

**Figure 4 pharmaceuticals-16-00856-f004:**
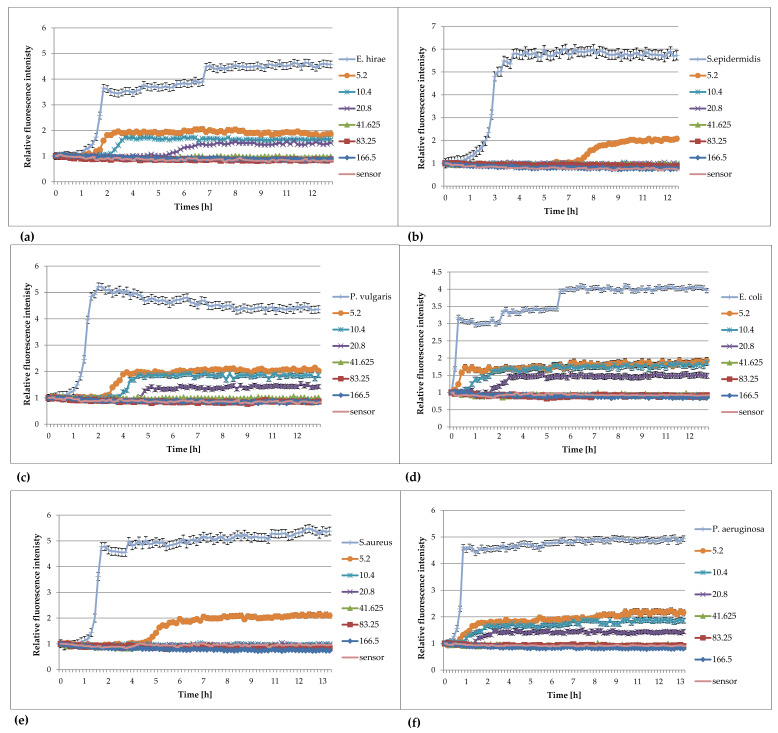
Plots of the relative fluorescence intensity of bacteria culture against time for different concentrations of extracts from *Lonicera caerulea* var. *edulis* ‘Wojtek’ fruits. Data presented in the legend contain concentrations of plant extracts (166, 83.25, 41.625, 20.8, 10.4, 5.2 mg/mL). (**a**) *Enterococcus hirae* ATCC 10541, (**b**) *Staphylococcus epidermidis* ATCC 14990, (**c**) *Proteus vulgaris* NCTC 4635, (**d**) *Escherichia coli* ATCC 8739, (**e**) *Staphylococcus aureus* ATCC 6538, (**f**) *Pseudomonas aeruginosa* ATCC 9027.

**Figure 5 pharmaceuticals-16-00856-f005:**
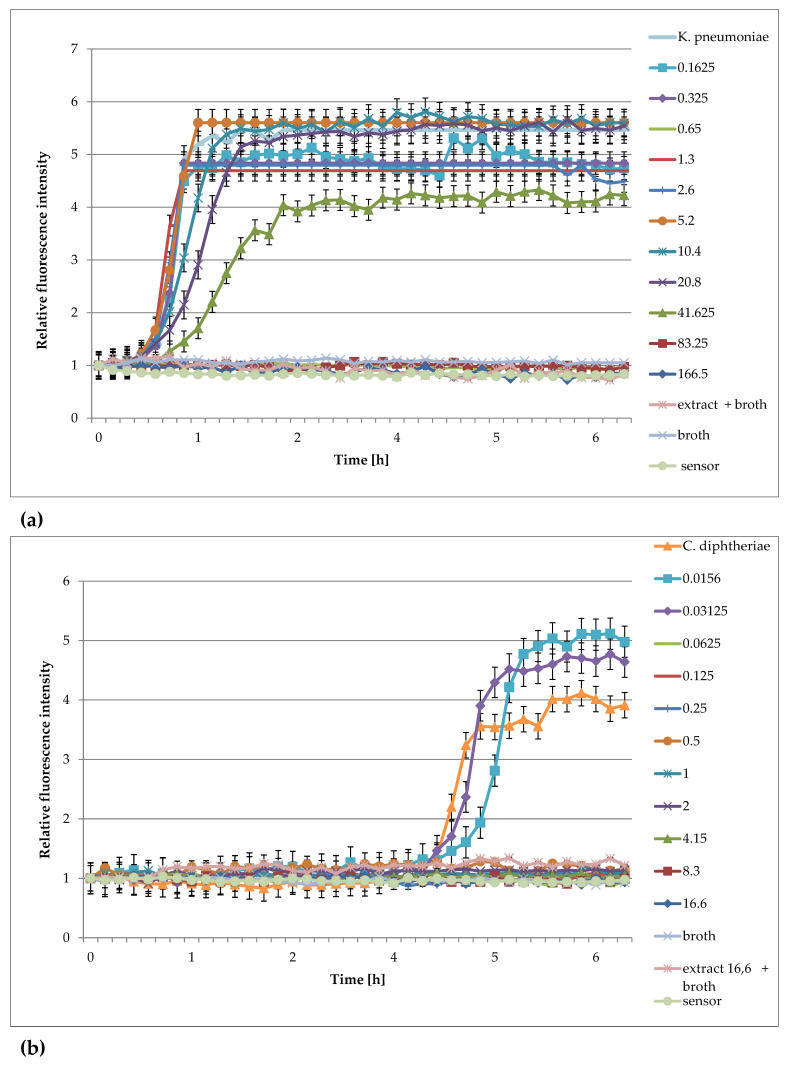
Plots of the relative fluorescence intensity of bacteria culture against time for different concentrations of extracts from *Rubus idaeus* shoots (166, 83.25, 41.625, 20.8, 10.4, 5.2 mg/mL or 16.6, 8.3, 4.15, 2.075, 1, 0.6 mg/mL). (**a**) *Klebsiella pneumoniae*, (**b**) *Corynebacterium diphtheriae*.

**Figure 6 pharmaceuticals-16-00856-f006:**
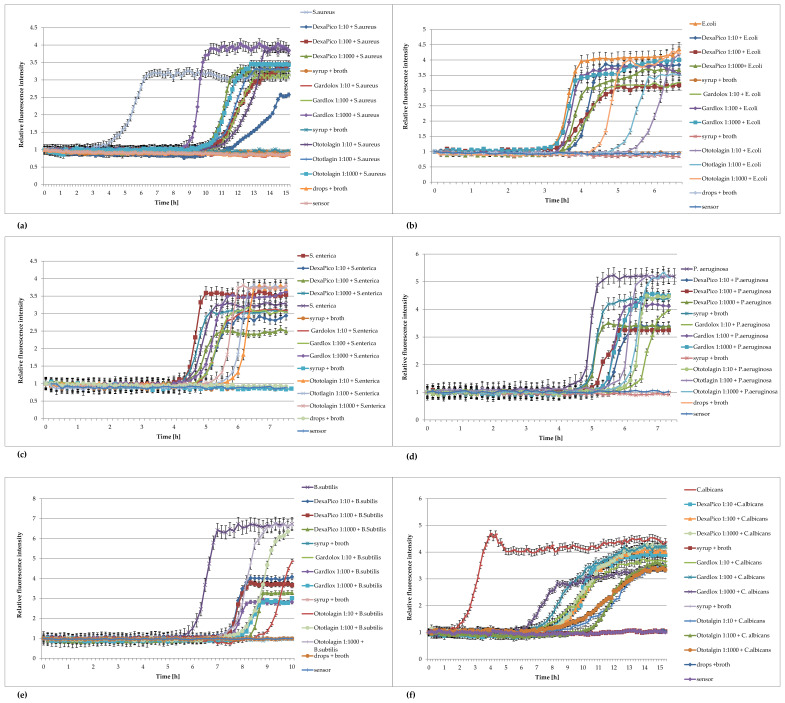
Plots of the relative fluorescence intensity of microorganism cultures against time for different concentrations of syrups: DexaPico, Gardlox, and the ear drops Ototalgin. Data presented in the legend contain dilution of non-sterile pharmaceutical products (1:10; 1:100; 1:1000). (**a**) *Staphylococcus aureus* ATCC 6538, (**b**) *Escherichia coli* ATCC 8739, (**c**) *Salmonella enterica* ATCC 13076, (**d**) *Pseudomonas aeruginosa* ATCC 9027, (**e**) *Bacillus subtilis* ATCC 6633, (**f**) *Candida albicans* ATCC 10231.

**Figure 7 pharmaceuticals-16-00856-f007:**
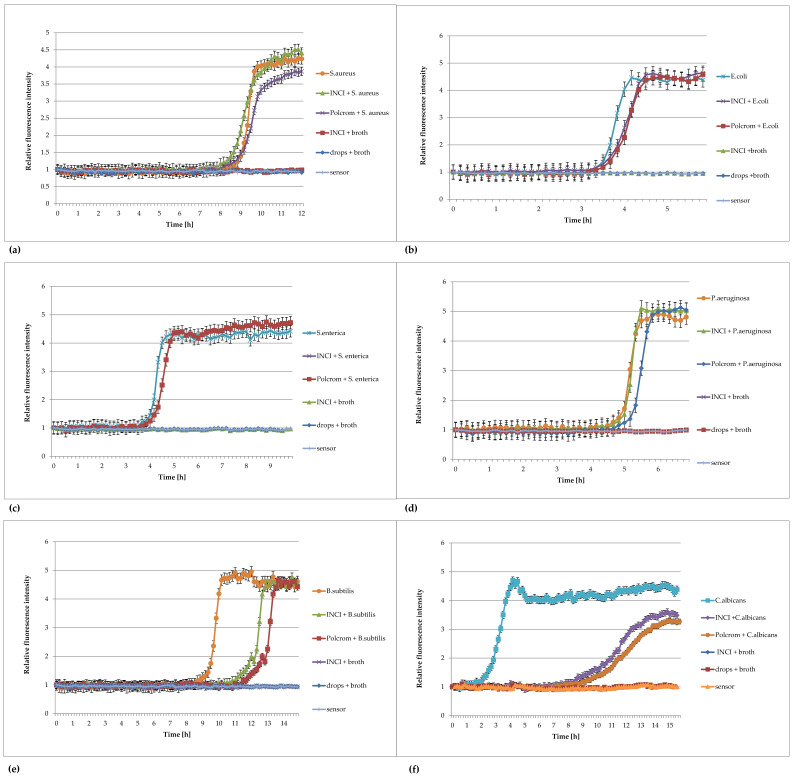
Plots of the relative fluorescence intensity of microorganism cultures against time for 1:10 dilution of Injection Natrii Chlorati Isotonica (INCI) and the eye drops Polcrom in culture. (**a**) *Staphylococcus aureus* ATCC 6538, (**b**) *Escherichia coli* ATCC8739, (**c**) *Salmonella enterica* ATCC 13076, (**d**) *Pseudomonas aeruginosa* ATCC 9027, (**e**) *Bacillus subtilis* ATCC 6633, (**f**) *Candida albicans* ATCC 10231.

## Data Availability

Not applicable.

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
