# Peer review of "Application of tris-(4,7-Diphenyl-1,10 phenanthroline)ruthenium(II) Dichloride to Detection of Microorganisms in Pharmaceutical Products†"

_pharmaceuticals, 2023, doi:10.3390/ph16060856_

Round 1

Reviewer 1 Report

The Manuscript by R. HaÅ‚asa, K. Turecka, M. Smoktunowicz, U. Mizerska, C. Orlewska “Application of tris-(4,7-diphenyl-1,10-phenanthroline)ruthenium(II) dichloride to detection of microorganisms in pharmaceutical products” describes application of fluorescent optical respirometry (FOR) method for the evaluation of viability of microorganisms in the presence of DMSO and plant extracts as well as the evaluation of content of aerobic bacteria in non-sterile pharmaceutical products.

Development of methods that allow fast and reliable estimating microbial contamination and content of viable microorganisms in pharmaceutical compositions, medicinal plant materials and food products represents an urgent task. FOR is the method that simplifies the analytic procedure of evaluation of viable microorganisms content. However its implementation requires the use of specific reagents, specific equipment and specific sample preparation. So, the accessability of reagents and equipment are likely to partially offset the benefits of the method.

Overall, the Manuscript is prepared well. However there are some issues that need to be improved prior to publication.

1.    Obviously, the FOR method secribed in the manuscript has some limitations and flaws. Application of this express method to new analytic samples is possible only after initial validation of its results by independent method (e.g. serial dilutions method in broth). It is necessary to discuss the limitations of the method and possible influence of components of pharmaceuticals or plant extracts on the results of FOR analysis in the manuscript.

2.    Authors themselves assert that reliability and reproducibility of analysis by FOR depends on the quality of preparation of sensor compound and its placing in the testing well (lines 161-164 in Manuscript). The method of sensor compound preparation, its placing in testing wells, its mixing with testing samples etc. have to be described in detail (in Materials and Methods section). The equipment used to measure the fluorescence should be specified as well.

3.    Graphs presented in the manuscript are poorly demonstrative. The multiple overlapping curves complicate the perception. I suggest to exclude some overlapping curves from Figures 1, 4 and 5. Figure 3 can be excluded at all or presented as a Table.

4.    Only one of abbreviations FRO or FOR should be used throughout the text.

5.    The list of references is should be formatted uniformly.

I believe that the Article from R. Hałasa, K. Turecka, M. Smoktunowicz et al. is of interest to a broad auditory of the Pharmaceuticals journal, but it can not be published in its present form. However, it deserves publishing after considering the comments above.

Author Response

Response to Reviewer 1 Comments

We would like to thank the Reviewer for reviewing our manuscript for their time and effort. We are grateful for all valuable and useful comments and remarks, their introduction to the manuscript will certainly contribute to its improvement. For document clarity, first, we quoted the Reviewer's comments and then our response in red, taking into account the page and line numbers of the changes made to the body of the manuscript. Moreover, the corrected places in the manuscript text were highlighted.

Point 1: Obviously, the FOR method described in the manuscript has some limitations and flaws. Application of this express method to new analytic samples is possible only after initial validation of its results by independent method (e.g. serial dilutions method in broth). It is necessary to discuss the limitations of the method and possible influence of components of pharmaceuticals or plant extracts on the results of FOR analysis in the manuscript.

Response 1: A limitation with minimal constraints is the availability of the sensor, a microplate reader that measures fluorescence, and the problem of distributing the sensor mixed with silicone to the wells.

The MIC results for the plant extracts are confirmed by serial dilution and published and referenced in the text lines 241 and 255. Only the MIC results for DMSO are unpublished but are verified as shown in figure 3 and mentioned in line 178.

Carrying out researche with pharmaceutical products, we used the validated METHOD SUITABILITY TEST contained in PhE.10.0, in monographs 2.6.1 and 2.6.12. We used appropriate media, dilutions, test strains, and research procedures, lines 283,367,504

The effect of the extract components on the test results and possible interferences were checked by incubation and measurements of the extracts themselves simultaneously with the actual samples. In all presented graphs, there are marked curves showing the measurements of the extracts themselves, visible in the form of flat lines close to one. This proves the lack of influence of the tested compounds on the results obtained from the proper tests. The situation is similar in the case of pharmaceutical products that were themselves incubated with the sensor. In lines 346-347 we mention that the growth of microorganisms is influenced by the preservatives used, which in our case results in a shift in the growth curves over time, i.e. it may delay the detection of microorganisms, but does not affect the detection of their presence. We recommend using appropriate neutralizers.

Point 2: Authors themselves assert that reliability and reproducibility of analysis by FOR depends on the quality of preparation of sensor compound and its placing in the testing well (lines 161-164 in Manuscript). The method of sensor compound preparation, its placing in testing wells, its mixing with testing samples etc. have to be described in detail (in Materials and Methods section). The equipment used to measure the fluorescence should be specified as well.

Response 2: the It has been done according to the Reviewer’s sugestion.

 Point 3: Graphs presented in the manuscript are poorly demonstrative. The multiple overlapping curves complicate the perception. I suggest to exclude some overlapping curves from Figures 1, 4 and 5. Figure 3 can be excluded at all or presented as a Table.

Response 3: We do not know exactly which curves you mean, most likely curves for control (negative) samples, sensor and samples, where the tested compounds inhibit the growth of microorganisms. In our opinion, these tests are an important element of researches, indicating the inhibitory values ​​and those in which microorganisms grow. Also, control tests in which we show the lack of interference of the extracts and pharmaceuticals on the results obtained are necessary.

In our opinion, Figure 3 is an attempt to show how the hundreds of obtained results presented in the separate graphs in Figure 1, can be used to demonstrate the results of the researches in clear and legible way.

Point 4: Only one of abbreviations FRO or FOR should be used throughout the text.

Response 4: According to the Reviewer’s comment the abbreviation FOR has been used in the text.

Point 5.    The list of references is should be formatted uniformly.

Response 5: According to the Reviewer’s suggestion the list of references have been corrected.

Reviewer 2 Report

The authors had published similar research about the FOR method. Although  the samples has been replaced,the manuscript is lack of novelty. A few suggestions are noted below:

1. Most of the references are not in the past five years.

2.The selected test samples have unclear classification and poor representativeness. It is difficult to explain the advantage and applicability of the method.

The manuscript needs language review.

Author Response

Response to Reviewer  2 Comments

We would like to thank the Reviewer for reviewing our manuscript for their time and effort. We are grateful for all valuable and useful comments and remarks, their introduction to the manuscript will certainly contribute to its improvement. For document clarity, first, we quoted the Reviewer's comments and then our response in red, taking into account the page and line numbers of the changes made to the body of the manuscript. Moreover, the corrected places in the manuscript text were highlighted.

Point 1: Most of the references are not in the past five years.

Response 1: The mentioned suggestion has been considered by the Authors in the revised version of the manuscript.

Point 2: The selected test samples have unclear classification and poor representativeness. It is difficult to explain the advantage and applicability of the method.

Response 2: We have selected pharmaceutical preparations available without a prescription on the market in Poland, whose basic ingredient is water, i.e. the substance that has the greatest impact on shortening the shelf life of products. Choosing the preparations, we were guided also by the multi-ingredient composition - plant extracts and syrups, which can have a multidirectional effect. We want to check whether we can measure such a composition containing aromatic organic compounds and inorganic compounds, using our sensor. In addition, we wanted to show that the signal creating during bacterial growth in the sample will not be disturbed by these compounds. Our goal was also to prove, that it is possible to test and detect the presence of microorganisms in such preparations in short real time. It was obtained for the first time in this work- the positive results for non-sterile and sterile products, where we could detect microbes without any problems in a short time. These results are the first one obtained using fluorescence optical respirometry method detecting medical products. We demonstrate the usefulness of the method as a quick and objective way to show the presence of microorganisms in the tested products. The presented results show the possibility of using FOR to test various medical products, regardless of their composition. Our further research will focus on checking other materials, using other sensors, as an example: Turecka, K.; Chylewska, A.; Dąbrowska, A.M.; Hałasa, R.; Orlewska, C.; Waleron, K. Ru(II) Oxygen Sensors for Co(III)Complexes and Amphotericin B Antifungal Activity Detection by Phosphorescence OpticalRespirometry. Int. J. Mol. Sci. 2023, 24, 8744. https://doi.org/10.3390/ijms24108744.

Reviewer 3 Report

pharmaceuticals-2389896

Comments:

Overall, the manuscript is well written. However, manuscript needs substantial revision in order to appreciate the quality before it can be considered for publication.

The abstract needs minor changes. It should state briefly the purpose of the research, the principal results and major conclusions. An abstract is often presented separately from the article, so it must be able to stand alone. Please restructure the abstract.

In the manuscript, there are spelling and grammatical errors. Grammatical and punctuation errors must be corrected. Examine the manuscript thoroughly.

Use full form along with abbreviation at first instance and in the rest of the manuscript abbreviations can be used. 

In material and methods section, minor changes are required. 

1. Abstract must improve by mentioning the significance of your study of the aim

2. Add more recent references in the Introduction section and focus on the objective of the study

4. Figures 4 & 7 must be improved for easily readable

5. Discussion part must be refined and add relevant work more

6. Conclusion part is not good to separate and indicates your goal of the study clearly

7. References should follow the journal style

8. Minor spelling mistakes, check it.

 9. Must improve the English language by editing with a native speaker

10. The background data need to be strengthened with recent findings and literatures. The authors have missed several recent papers on the study. 

11. Merit of the manuscript is below:

Impact of the article: Top 60%

 Originality of article: Top 50%

Quality of data and methods: Bottom 60%

 Standard of English: Top 55%

My Opinion: Major Revision

Author Response

Response to Reviewer 3 Comments

We would like to thank the Reviewer for reviewing our manuscript for their time and effort. We are grateful for all valuable and useful comments and remarks, their introduction to the manuscript will certainly contribute to its improvement. For document clarity, first, we quoted the Reviewer's comments and then our response in red, taking into account the page and line numbers of the changes made to the body of the manuscript. Moreover, the corrected places in the manuscript text were highlighted.

Point 1: Abstract must improve by mentioning the significance of your study of the aim

Response 1: According to the Reviewer’s suggestion the abstract has been corrected.

Point 2: Add more recent references in the Introduction section and focus on the objective of the study

Response 2: More recent references in the Itroduction have been added, according to the Reviewer’s suggestion; lines 127-130.

Point 4:  Figures 4 & 7 must be improved for easily readable

Response 4: The mentioned suggestion has been considered by the Authors in the revised version of the manuscript. New versions of the figures have been added.

Point 5:  Discussion part must be refined and add relevant work more

Response 5: The lack of literature describing the similar sensors in pharmacy makes it impossible to discuss the results obtained

Point 6:  Conclusion part is not good to separate and indicates your goal of the study clearly

Response 6: The mentioned suggestion has been considered by the Authors in the revised version of the manuscript.

Point 7:   References should follow the journal style

Response 7: According to the Reviewer’s suggestion the References have been corrected.

Point 8: Minor spelling mistakes, check it.

Response 8: The mistakes have been corrected.

 Point 9: Must improve the English language by editing with a native speaker

Response 9: According to the Reviewer’s suggestion the English language has been improved.

Point 10:  The background data need to be strengthened with recent findings and literatures. The authors have missed several recent papers on the study. 

Response 10: The mentioned suggestion has been considered by the Authors in the revised version of the manuscript.

Round 2

Reviewer 2 Report

Accept